# Revised historical Northern Hemisphere black carbon emissions based on inverse modeling of ice core records

Sabine Eckhardt [1] ✉, Ignacio Pisso[1], Nikolaos Evangeliou [1], Christine Groot Zwaaftink[1], Andreas Plach [2], Joseph R. McConnell [3], Michael Sigl [4,5], Meri Ruppel[6,7], Christian Zdanowicz [8], Saehee Lim [9], Nathan Chellman[3], Thomas Opel [10], Hanno Meyer [10], Jørgen Peder Steffensen [11], Margit Schwikowski [12] & Andreas Stohl [2]

Black carbon emitted by incomplete combustion of fossil fuels and biomass has a net warming effect in the atmosphere and reduces the albedo when deposited on ice and snow; accurate knowledge of past emissions is essential to quantify and model associated global climate forcing. Although bottom-up inventories provide historical Black Carbon emission estimates that are widely used in Earth System Models, they are poorly constrained by observations prior to the late 20th century. Here we use an objective inversion technique based on detailed atmospheric transport and deposition modeling to reconstruct 1850 to 2000 emissions from thirteen Northern Hemisphere ice-core records. We find substantial discrepancies between reconstructed Black Carbon emissions and existing bottom-up inventories which do not fully capture the complex spatial-temporal emission patterns. Our findings imply changes to existing historical Black Carbon radiative forcing estimates are necessary, with potential implications for observation-constrained climate sensitivity.

Black carbon (BC) is a fraction of carbonaceous aerosol that has a strong positive radiative forcing and can result in severe health impacts at high concentrations. BC is considered to be among the most important climate forcers at the global scale together with carbon dioxide and methane[1,2], with a present warming contribution estimated to be 0.1 degree relative to pre-industrial times, according to the recent Sixth Assessment report by the Intergovernmental Panel on Climate Change[3]. Its climate effect is amplified when BC is deposited in snow- and ice-covered areas resulting in decreased albedo and increased snow melt[3–6]. Because of its relatively short atmospheric

lifetime (days to weeks), BC is heterogeneously distributed in the atmosphere so the lack of global observations capable of capturing this heterogeneity means that modelling is crucial to assess atmospheric concentrations. The potential for climate change mitigation through BC emissions reductions has been recognized. For the development of successful mitigation strategies accurate historical BC emissions are crucial. However, many bottom-up emission inventories are poorly constrained by observations[1].

Global BC emissions are highly variable and depend on regional emission sources and policies. South Asia currently suffers the world's

[1]NILU - Norwegian Institute for Air Research, Kjeller, Norway. [2]Department of Meteorology and Geophysics, University of Vienna, Vienna, Austria. [3]Division of Hydrologic Sciences, Desert Research Institute, Reno, NV 89512, USA. [4]Environmental Physics, Physics Institute, University of Bern, 3012 Bern, Switzerland. [5]Oeschger Centre for Climate Change Research, University of Bern, 3012 Bern, Switzerland. [6]Atmospheric Composition Unit, Finnish Meteorological Institute, Helsinki, Finland. [7]Ecosystems and Environment Research Programme, University of Helsinki, Helsinki, Finland. [8]Department of Earth Sciences, Uppsala University, Uppsala, Sweden. [9]Department of Environmental Engineering, Chungnam National University, Daejeon 34134, South Korea. [10]Alfred Wegener Institute Helmholtz Centre for Polar and Marine Research, Potsdam, Germany. [11]Niels Bohr institute, University of Copenhagen, Copenhagen, Danmark. [12]Paul Scherrer Institut, Villigen, Switzerland. ✉e-mail: sec@nilu.no

highest BC pollution levels, which not only perturb Earth's radiative balance, but also threaten human health by causing premature mortality[7]. Conversely, BC pollution in North America and Europe has been decreasing in the last decades[8,9]. These emission changes are directly linked via long-range atmospheric transport to BC concentrations and deposition rates in the Arctic. Several Arctic air monitoring stations that have been operating since 1989[10] show decreasing atmospheric BC trends during recent years[11,12].

Long-term constraints on past BC emissions beyond the atmospheric monitoring era can be reconstructed from natural archives incorporated in lake sediments and glacier ice. While lake sediments are important archives of BC deposition since they are distributed throughout terrestrial regions, interpretation is hampered by low resolution, generally high dating uncertainties, and difficulties in quantitatively relating sediment records to atmospheric concentrations and emissions, in part because BC transport to lake sediment sites can be both aeolian and fluvial[13,14]. Contrary to sediment records, ice cores from the Greenland ice sheet and smaller mid- and high-latitude mountain glaciers and ice caps provide direct, well-resolved, and well-dated records of solely atmospherically deposited BC. Similar to recent studies quantifying ancient, Medieval, and modern atmospheric lead[15] and other pollutant emissions[16], ice-core records provide robust constraints on past BC emissions[15,17,18].

The first high-resolution ice-core record of BC concentrations[19] was collected from west-central Greenland covering the period 1790–2005. It showed a large influence of BC from boreal forest fires before 1890 (hereafter referred to as biomass burning (BB) BC), and that anthropogenic BC contribution peaked in the 1920s before decreasing during the last few decades of the 20th century. Since this record was published, several other ice cores have been developed from Greenland[20–25], Arctic Canada[26], Svalbard[27,28], the continental United States[29], Russia[30], the European Alps[31,32] South America[33] the Antarctic[17,34,35], the Himalayas[36] and the Tibetan Plateau[37]. These records show high interannual variability in large part because of year-to-year variability in transport and seasonal snow deposition at the coring sites, but robust multi-annual trends related to historical changes in regional emissions are clear.

Interpretation of some of these records has been underpinned by detailed atmospheric transport and deposition modeling[17,35]. Bauer et al., 2013[25] reported underestimation of modelled BC deposition for Greenland at the beginning of the 20th century and overestimation toward the end of the century. Comparisons of simulated BC deposition in aerosol transport models[38,39] with snow/ice observations revealed biases likely related to deficiencies in the underlying emission inventories. In the Arctic, significant uncertainties in historical emission inventories are related to inaccuracies in the spatial representation and high variability in emission estimates for individual source sectors[2]. High uncertainties in BC emissions are directly propagated to modelled atmospheric transport and deposition, and consequently to climate impact assessments of BC[1]. Therefore, more accurate quantification of BC emissions is fundamental for climate modelling. It is crucial to provide independent constraints on BC emissions, especially for historical emissions that are associated with the largest uncertainties because of a lack of observations. Here we use a formal inversion technique to quantitatively constrain the emissions in several broad regions based on a large array of ice-core BC records. While earlier studies have used individual ice cores to discuss implications for historical emissions[25], here we make use of an array of thirteen BC ice-core records collected from mid- to high-latitude Northern Hemispheric sites to constrain historical BC emissions since the late 19th century Industrial Revolution. BC deposition records were collected from Greenland, Arctic Canada, Russia, Svalbard, and mid-latitude glaciers in the European Alps and the Caucasus (supplementary Table S1). All the ice cores, except the Holtedahlfonna record, were analyzed using the

single-particle soot-photometer method[19] so BC concentrations should be closely comparable, although there were differences in sample handling (e.g., continuous versus discrete sample measurements[40] that may have affected some of the records. Temporal resolutions vary from sub-annual (Greenland) to sub-decadal (e.g., Canada); hence, we here focus on trends in decadal-scale deposition so the effects of any dating uncertainties, sampling resolutions, or post-depositional processes should be relatively small[18]. The unprecedented size and geographic breadth of this ice-core array allows for exceptional robustness in the inferred BC emissions since BC deposition trends recorded in multiple ice cores likely reflect changes at regional to continental scales.

For each ice-core site we calculated monthly footprint emission sensitivities (FES) that allow quantification of the impact of emissions on the BC deposition at the receptor. FES is the probability that emissions occurring at any grid cell lead to deposition at the receptor. Thus, FES can be high even in regions where there are no BC emissions.

We used meteorological re-analysis data for the 20th century (Methods) for the calculations of the FES and extended our calculations back to 1850 assuming that dynamics of atmospheric transport and deposition in the 19th century were the same as in the 20th. Similar calculations have been performed to investigate historical deposition of lead in Greenland[41] and BC in Antarctica[17]. Finally, the modelled FES together with the thirteen ice core observations of BC were used in an inversion framework to quantitatively constrain emissions in several broad regions (Methods).

## Results and discussion
### Source regions for the ice core locations
Results show that the northern Greenland high-altitude ice-core sites (Fig. 1a) have lower FES values overall and particularly in the Arctic, reflecting the fact that air arrives at these high-altitude sites (≥2000 m a.s.l.) via the free troposphere and has less surface contact than air arriving at lower-altitude sites. Southern Greenland ice core sites (Fig. 1b) are sensitive to BC emissions in North America but also in the UK and Scandinavia. The low-altitude Flade Isblink site in northeastern Greenland (Fig. 1d), Devon ice cap in Canada, and the three sites in the European Arctic (Fig. 1c) are more sensitive to emissions in Eurasia than to North America. In particular, the Akademii Nauk and Svalbard (Fig. 1c) ice core sites are highly sensitive to emissions in Russia and Northern Europe. The Colle Gnifetti (Fig. 1f) and Mt. Elbrus sites (Fig. 1g) are mostly sensitive to emissions in Western and Central Europe, and Eastern Europe and Russia, respectively. These results show that the different ice cores receive BC emitted from different source areas, but also that their FES overlap, implying that no ice core record is uniquely representative of BC emissions from only one region. However, collectively they provide regional emission information that can be extracted from the joint record.

### Modelled and observed black carbon deposition fluxes
By combining the FES (Fig. 1) with different emission inventories we obtained modeled BC deposition fluxes at the ice core sites. We did this separately for two BC emission inventories used for the 5th and 6th climate model inter comparison projects (CMIP5 and CMIP6; Fig. S1), which can be directly compared to observations. However, to avoid biases (details below) when performing the inversion, we scaled the modelled concentration to the observations; the original/unscaled values can be found in the supplement in Fig. S3. Mean observed BC deposition fluxes (Fig. 2) at the various ice-core sites over the period 1850–2000 span an order of magnitude, with the lowest values at high altitude Greenland sites and Devon (0.2–0.3 mg m$^{-2}$ year) and the highest values at Mt. Elbrus (4.2 mg m$^{-2}$ year) and Holtedahlfonna, Svalbard (12.3 mg m$^{-2}$ year). However, the higher flux at Holtedahlfonna is most likely due to a different BC quantification method used at this site, which is known to produce higher concentrations

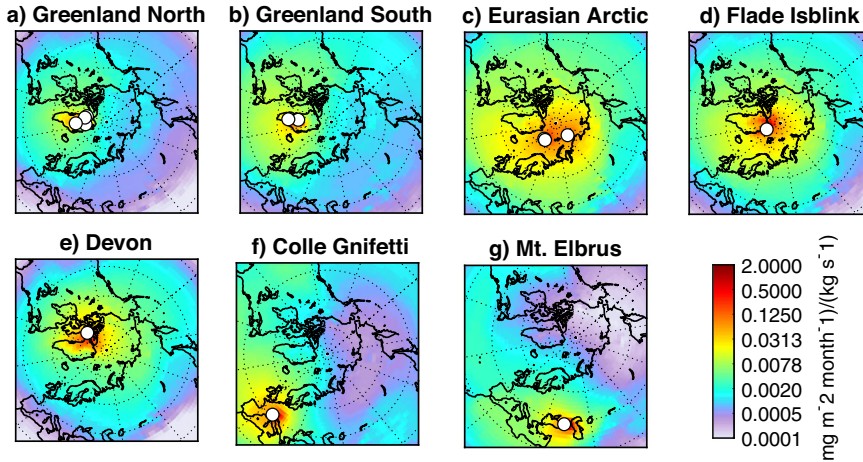

**Fig. 1 | Sensitivity of black carbon deposition at Northern Hemisphere ice core sites to source region emissions.** The footprint emission sensitivity (FES) in a grid cell is the simulated black carbon deposition (mg BC m$^{-2}$ month$^{-1}$) at the ice core site that a potential emission source of unit strength (1 kg s$^{-1}$) in that grid cell would produce[57]. It combines wet and dry black carbon deposition, and accounts for black carbon source emissions below 100 m a.g.l., averaged over 100 years. The ice core sites are marked with white dots. For ice core sites close to each other, FES values are averaged. We combine ice cores in the region of northern Greenland (**a** - Summit, Tunu, NEEM, Humboldt), southern Greenland (**b** - ACT2, D4), three sites in the Eurasian Arctic (**c** - Holtedahlfonna, Lomonosovfonna and Akademii Nauk), and show separately a low-altitude Greenland site (**d** - Flade Isblink), one in the Canadian Arctic (**e** - Devon), and in the mid-latitudes Colle Gnifetti in the Alps (**f**) and Mt. Elbrus in the Caucasus (**g**). Notice that emission sensitivity maps are combined here for display purposes only but are treated separately in the analysis.

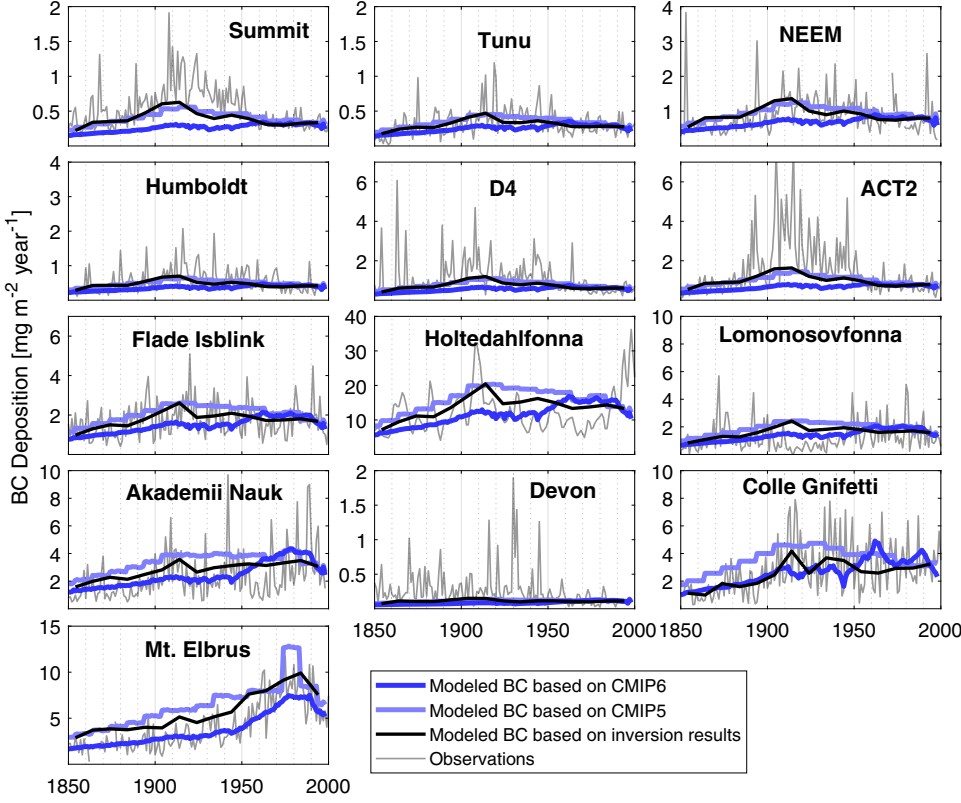

**Fig. 2 | Comparison of measured and modeled black carbon (BC) deposition fluxes for the thirteen ice cores.** Observed (decadal) mean annual black carbon deposition fluxes (gray) at all sites except for Holtedahlfonna, for which elemental carbon (EC) deposition is given with a sampling resolution of ca. 2–5 years. The a priori modeled values were obtained by using the biomass burning emissions and the black carbon inventories for CMIP5 (light blue) and CMIP6 (dark blue), combined with monthly emission sensitivities and scaled to the measured fluxes of the latest available decade. Modeled BC deposition fluxes using *a posteriori* emissions from the inversion are shown with black lines. Notice that the ordinate scales are different in each panel.

(Supplementary Data Table S1), and local contamination from Svalbard coal mines has been ruled out by several previous studies[27,42]. Generally, these observed differences in BC deposition fluxes between the different ice cores of about one order of magnitude are reflected well in the modeled fluxes which span from 0.9 mg m$^{-2}$ year for Tunu to 7.0 mg m$^{-2}$ year for Mt. Elbrus when using CMIP5 emissions and 0.6 mg m$^{-2}$ year to 4.1 mg m$^{-2}$ year with CMIP6 emissions (Fig. S3), although substantial biases exist at individual sites.

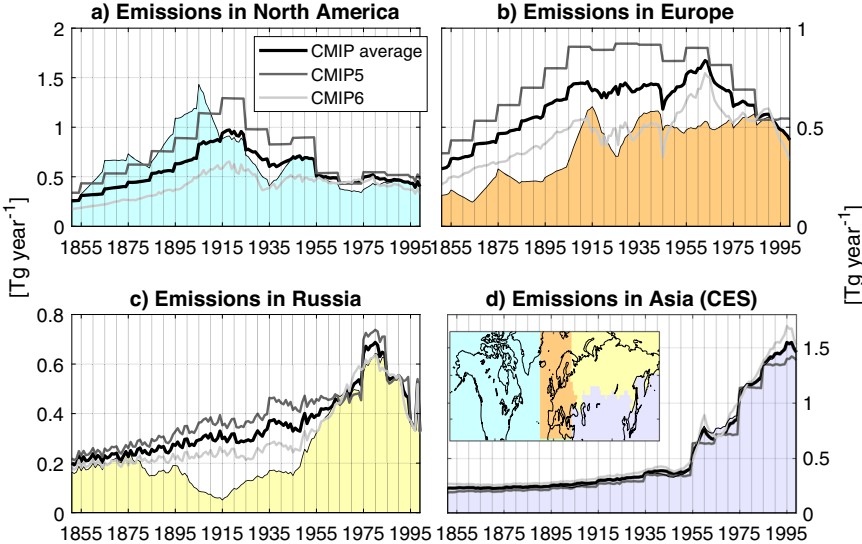

**Fig. 3 | Black carbon emission history obtained from the inversion.** A priori (thick black line) and *a posteriori* (shaded areas) emissions north of 30 °N for the different regions (**a**–**d**). CMIP5 and CMIP6 emissions (both including the same biomass burning emissions) are shown with dark and light gray lines, respectively. The regions for which the emissions are optimized are shown as an inset in (**d**) and reflect the colors used in (**a**–**d**).

In most Greenland ice-core records observed BC deposition peaks around 1910 and then decreases toward 2000 (Fig. 2). At ACT2, D4 and Summit, post-1950 fluxes are about 25% of early 20th century values, with no prominent later increases. The timing of the early 20th century BC deposition maximum in Greenland ice cores is much better captured by our model when using CMIP5 rather than CMIP6 emissions, suggesting that early 20th century BC emissions from North America are incorrectly represented in the CMIP6 inventory. In contrast to Greenland, the Akademii Nauk and Holtedahlfonna ice cores (Eurasian Arctic) and the Mt Elbrus core show rising BC deposition in the late 20th century, peaking in the 1970s on Mt Elbrus and the 1980s at Akademii Nauk and Holtedahlfonna. This is consistent with continuous increases in continental BC emissions at lower latitudes in Russia and Central, East and South (CES) Asia, documented in both the CMIP5 and CMIP6 inventories, and which only ended in the 1980s (Fig. S1). Increasing BC deposition through the late 20th century is also documented in lake sediments from north-western Arctic Russia. This was attributed in part to growing BC emissions from natural gas flaring in the mainland Russian Arctic[14] which is an important source region for Akademii Nauk and Holtedahlfonna (Fig. 1). Greenland ice core records (ACT2, Summit, D4 and Tunu) have comparatively low BC deposition fluxes in the late 20th century, suggesting that the 1980s maximum in the Akademii Nauk and Holtedahlfonna ice cores is mostly due to BC emissions from the former Soviet Union, which dropped in the 1980s because of the economic collapse and political disintegration. The temporal evolution of BC emissions in the former Soviet Union is likely better represented in the CMIP6 than the CMIP5 inventories, since there is a better agreement between the modeled and measured BC deposition fluxes in the second half of the 20th century at Akademii Nauk, Holtedahlfonna and Mt. Elbrus (Fig. 2) when using the CMIP6 data.

**Inversion results**

The fact that different ice-core sites are sensitive to BC emissions from different source regions (Fig. 1) provides an opportunity to use inverse modelling for constraining historical emissions with the ice-core records. This approach requires the use of so-called a priori emissions, for which we use the average of CMIP5 and CMIP6 inventories, combined with BC emission data from biomass burning (BB[43]; for emission variation see supplementary data Fig. S1). We then applied a Bayesian inversion algorithm (Methods) to the decadally-averaged ice-core records of BC deposition from 1850 to 2000 to determine optimized *a posteriori* BC emissions in four source regions (North America, Europe, Russia and the CES Asia; Fig. 3d, inset). A potential problem with this approach is that model biases or inaccuracies in the ice-core records (e.g., age model, post-depositional snow scouring by wind or melt water percolation) can be imprinted on the emissions retrieved by the inversion. To bypass this problem, we scaled the modeled BC deposition fluxes for each ice core to the observed values of the last decade (Supplementary Data Fig. S2), on the assumption that the a priori *emissions* are most reliable during that period, and that model and observational biases and the representativity of the sites did not change with time. This emphasizes temporal variability and removes uncertainties associated with absolute BC concentrations measured using different methods. Sensitivity studies without scaling the model results to observations, show that European emissions are unrealistically low, while the other emissions are unaffected (Supplementary Data Fig. S9).

The largest change between a priori (inventory-based) and *a posteriori* (updated after the inversion) BC emissions is an increase of the emissions in North America from 1880 to 1920 (Fig. 3a). Inverse modelling suggests that BC emissions there peaked higher and earlier (in the 1900s) than reported in CMIP emission inventories. Following this peak, *a posteriori* emissions drop rapidly to a minimum during the Great Depression of the 1930s[44] when they reach about one third of their 1900s maximum. This rapid drop in emissions is also absent in the CMIP inventories. Modeled emissions in the second half of the 20th century are more consistent with the inventories. Sensitivity tests showed that the results obtained for North America are robust against changes in the inversion setup (Supplementary Data Figs. S9 and S10).

For Europe, the inversion decreases the a priori emissions over the whole period, with the exception of the late 20th century, when the emissions are nearly unchanged (Fig. 3b). The largest difference is that the a priori emissions start to decrease in the early 1970s, whereas the *a posteriori* emissions remain stable until the late 1980s, and then decline. For Russia, the modeled *a posteriori* emissions are substantially lower than those in the CMIP inventories from about 1900 to 1950 (Fig. 3c). The largest difference is in the 1910s, when posterior emissions are up to 65% lower. It seems plausible that Russian BC emissions decreased during this time, a consequence of the turmoils of the October Revolution and World War I, which caused the economy to collapse[45,46]. The early 20th century drop in a posteriori Russian BC

emissions is entirely absent in the CMIP5 and CMIP6 emission inventories, which show a continuous increase of BC emissions in Russia, with little variability. Before 1900 and after 1950, a priori and *a posteriori* BC emissions are essentially identical, and clearly show the steep decline in Russian BC emissions during the 1980s.

For CES Asia, a priori and *a posteriori* BC emissions are nearly the same (Fig. 3d). This is likely a result of the limited sensitivity of BC deposition at our 13 ice-core sites to emissions from this region. This limitation could be overcome in the future by including ice-core records of BC from central/northern Asia in the inversion.

Our results, which are based on an exceptionally robust and geographically representative ice core BC deposition dataset, should be implemented to improve used emission inventory data. We were able to define amendments required to emission inventories in specific areas of the Northern Hemisphere. Our findings have implications for estimates of global and regional radiative forcing caused by BC emissions in the last 150 years. Specifically, early 20th century BC emissions in Europe and Russia are overestimated by the existing emission inventories, whereas BC emissions in North America are underestimated. For North America, our modeled *posterior* emissions are about a factor two higher than the CMIP6 emissions for the period 1850–1920, and nearly a factor three higher in the 1900s. It is thus likely that climate models using the CMIP inventories strongly underestimate the BC radiative forcing in the western hemisphere but strongly overestimate it in the eastern hemisphere, possibly affecting global circulation. These discrepancies would affect simulated surface temperatures and thus also constraints on simulated climate sensitivity[47], with consequent implications for projections of future warming. Thus, future efforts should be directed to further refining existing BC emissions inventories for the early 20th century to increase the reliability of climate models. Increased emissions in the early 20th century might also have impacted melting of Arctic ice due to albedo changes. Since our inversions suggest similar emissions during the Industrial Revolution and in the recent past, a higher melting rate seems possible compared to what climate models currently assume.

## Methods

BC is a highly variable mixture of carbonaceous compounds with five underlying physical properties relating to its microstructure, morphology, thermal stability, solubility and light absorption[48]. Currently, there is no single instrument that can measure all properties simultaneously and no unique, standard method has been developed for BC quantification. All but one of the 13 ice core records used in this study were developed using the Single Particle Soot Photometer (SP2[49]), which is sensitive to refractory, sub-micron-diameter BC particles (refractory BC, or rBC). In short, ice meltwater samples are aerosolized using a nebulizer and introduced to the SP2, which measures individual BC particles using laser-induced incandescence to obtain a mass concentration. This approach can be used for both continuous ice core analysis[19,17] or for discrete meltwater samples[28]. The Holtedahlfonna ice core used in this study was analyzed discretely using a thermal-optical method[50] which detects a wider range of BC compounds than the SP2 method. Table S1 summarizes the ice core records and respective measurement methods.

Potential sources of uncertainties in the BC deposition fluxes include instrumental limitations (e.g., under-detection of very large BC particles), dating errors (<±5 years in some cores), post-depositional effects (e.g., wind scouring of snow, melt-refreeze cycles) and spatial noise (variability in deposition across space). For cores collected in the dry firn zone of Greenland, these sources of uncertainties are relatively minor, but some are poorly quantified at other coring sites, and will require further study. However, it is unlikely that such uncertainties

would affect BC deposition trends on decadal scales that are the focus of the present study.

For each ice core, following standard procedures, the annual BC deposition flux (mg m$^{-2}$ yr$^{-1}$) was calculated by multiplying the observed BC concentrations by the estimated annual net water-equivalent accumulation. For Holtedahlfonna, EC deposition was calculated by dividing the total amount of EC on a filter sample by the known cross section of the ice sample and the number of years covered in one filtered ice sample[27].

### FLEXPART modeling

As in previous studies interpreting ice core data[15,17,41], the Lagrangian particle dispersion model FLEXPART version 10.4[51,52] was used to obtain FES for the ice cores. The FLEXPART runs ingested meteorological input from the coupled climate reanalysis for the 20th century[53,54] (CERA-20C) of the European Centre for Medium Range Weather Forecasts (ECMWF) at a resolution of 2° x 2°, every 6 h, and with 91 vertical levels. For the time period of 1900–1999 we performed monthly backward simulations from each ice core location, using FLEXPART's capacity to quantify FES of the deposition in backward mode[55]. As the height of the ice core locations at some mountain sites (especially Colle Gnifetti and Mt. Elbrus) is severely underestimated in the CERA-20C data (Table S1), the scavenging height considered for wet deposition was corrected to the real height of the topography for all locations. The definition of the BC aerosol is taken from ref. [56]. Similar to the widely used backward mode for atmospheric concentrations[55], we obtain spatially resolved FES field[57] s on a three-dimensional grid. They represent the deposition at the receptor (μg m$^{-2}$ s) that would result from an unit emission of 1 kg s$^{-1}$ in a respective model grid cell. For the analysis we used monthly resolved footprints of the 100 years averaged over the 100-year CERA-20C period, combined with emissions from 1850–2000 to get modelled deposition fluxes.

### Emissions

For biomass burning BC emissions, we used historical emissions (1750–2015)[58]. For anthropogenic BC emissions, we used two different datasets. One is the historical gridded anthropogenic emission inventory[59] used for the Coupled Model Inter comparison Project (CMIP) 5[60], and the other the more recent inventory[61] used for CMIP6[62]. Both inventories provide monthly varying emission fields covering the periods from 1850–2000 to 1750–2014, respectively, on a 0.25 degree resolution. More details on the regional distribution of the two emission datasets can be found in the supplementary material.

### Inverse modeling

To reconstruct historic emissions in four regions (North America, Europe, Russia and CES Asia), we used an inversion method based on a maximum a posteriori (MAP) estimate with Gaussian errors. The transfer matrix was built using modeled decadal average deposition fluxes at each site based on FLEXPART simulations. For each of the fifteen decades (1850–2000), the modeled deposition contributed by each emission region was rearranged into a sparse matrix where emissions were considered independent of each other. The fifteen decadal values for each of the thirteen sites resulted in an observation vector **y** of length 195. The source regions were aggregated into four regions resulting in a control vector **x** of length 60. The corresponding transfer matrix **H** has therefore 195 rows and 60 columns. The prior emission factors $x_0$ (a vector of ones of length 60) multiplied by **H** yield the modeled deposition values. The cost function is

$$J(\mathbf{x}) = (\mathbf{y} - \mathbf{Hx})\mathbf{R}^{-1}(\mathbf{y} - \mathbf{Hx}) + (\mathbf{x_0} - \mathbf{x})\mathbf{B_0}^{-1}(\mathbf{x_0} - \mathbf{x}) \qquad (1)$$

where $\mathbf{R}$ and $\mathbf{B_0}$ are the error covariance matrices of the observations and the emission factors, respectively[63]; with the notation of [63,64]. In both cases the off-diagonal terms were set to zero. The diagonal terms of $\mathbf{R}$ contain contributions from the instrumental error of the observations $\mathbf{R_o}$ and transport model uncertainty $\mathbf{R_m}$ (we neglect other terms such as the representation error due to lack of information). The diagonal terms of $\mathbf{R_o}$ are set to the mean value of the observations from an assumed instrumental error of a factor two between different instrumental techniques from the literature[65]. The diagonal terms of $\mathbf{R_m}$ are the decadal variances of the observed values (that are affected by variability in transport). The diagonal terms of $\mathbf{B_0}$ were chosen based on the magnitude of the relative difference between the CMIP5 and CMIP6 emission and increase backward in time based on the assumption that earlier emissions are more uncertain than more recent ones. Therefore, the relative error is 30% for the most recent decade, linearly increasing to 100% for the first decade (1850–1860). The resulting posterior[63] emission factors are

$$\mathbf{x} = \mathbf{x_0} + \mathbf{B_0}\,\mathbf{H^T}(\mathbf{H}\mathbf{B_0}\,\mathbf{H^T} + \mathbf{R})(\mathbf{y} - \mathbf{H}\mathbf{x_0}) \qquad (2)$$

## Data availability

Links to emission inventories: https://esgf-node.llnl.gov/search/input4mips/ (van Marle et al., 2016). https://esgf-node.llnl.gov/search/input4mips/ (Hoesly et al., 2018). Meteorological data are taken from the CERA-20 project, for more details www.ecmwf.int. Ice core data can be accessed via: Flade Isblink, Akademii Nauk: https://doi.org/10.18739/A2W37KX21, Humboldt, https://arcticdata.io/catalog/view/doi:10.18739/A23T9D689, Tunu https://arcticdata.io/catalog/view/doi:10.18739/A2ZQ1G; Neem: Sigl, Michael; McConnell, Joseph R (2022): NEEM-2011-S1 ice-core aerosol record (conductivity, NH4, NO3, BC, acidity, Na, Mg, S, Ca, Mn, Sr, Ce) in NW-Greenland at 2 cm resolution from 86 to 1997 CE on the annual-layer counted NS1-2011 chronology. PANGAEA, https://doi.org/10.1594/PANGAEA.940553; Colle Gnifetti: Sigl, Michael; Abram, Nerilie J; Gabrieli, Jacopo; Jenk, Theo M; Osmont, Dimitri; Schwikowski, Margit (2018): Annual record of black carbon (rBC), bismuth, lead and others from 1741 to 2015 AD from Colle Gnifetti ice core (Swiss/Italian Alps). PANGAEA, https://doi.org/10.1594/PANGAEA.894785,; Mt. Elbrus: https://doi.org/10.5194/acp-17-3489-2017; The rBC data from the Devon ice cap in ASCII text format through the Canadian Polar Data Catalogue: ttps://www.polardata.ca/pdcsearch/PDCSearch.jsp?doi_id=12952 All figures were created using matlabV20b. The post-processing codes can be obtained from the corresponding author upon request.

## Code availability

The Lagrangian particle dispersion model FLEXPART is open access and FLEXPART version 10.4 is available at www.flexpart.eu. The Control files used for creating the output, as well as the average emission sensitivities used for the analysis can be downloaded here: https://folk.nilu.no/~sabine/BCICECORE/.

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

## Acknowledgements

We acknowledge ECMWF for the meteorological data. Eight of the thirteen ice-core BC records used in this study were developed at the Desert Research Institute (DRI), with collection and measurements supported by the U.S. National Science Foundations (including grants 0221515, 0909541, 1023672), as well as ongoing interpretation by grants 1925417 and 2102917. Drilling of the Akademii Nauk and Flade Isblink cores was led by the Alfred Wegener Institute and University of Copenhagen, respectively, and samples provided to DRI for analysis. The Devon ice cap core was drilled by the Geological Survey of Canada and analyzed with support from Curtin University and an Australian Endeavour Research Fellowship. We thank the field teams for collection of the cores and the laboratory students and staff who analyzed them. For Mt. Elbrus data, the PEGASOS project funded by the European Commission under the Framework Programme 7 (FP7-ENV-2010-265148).

## Author contributions

S.E. and A.S. designed the study, S.E. performed the analysis, N.E., A.P., and C.G. assisted with the FLEXPART modelling. I.P. performed the inversion. J.M., M.S., M.R., C.Z., S.L., N.C., T,O., H.M., J.S., and M.S. contributed ice-core black carbon datasets. All authors contributed to interpretation, discussion and writing of the results.

## Competing interests

The authors declare no competing interests.
