## [Peer Review File · Nature Communications]

Revised historical black carbon emissions based on inverse modeling of ice core recordsREVIEWER COMMENTS

Reviewer #1 (Remarks to the Author):

Review of “Revised historical black carbon emissions based on inverse modeling of ice core records”

This work analyzes BC records from 13 ice cores from the Arctic, Europe and Asia and uses an inversion technique based on atmospheric transport and deposition modeling to reconstruct past BC emissions from North America, Europe, Russia and Asia. The results disagree with existing bottom-up inventories of BC emissions and imply that climate models using the CMIP5 and CMIP6 inventories underestimate BC radiative forcings in the western hemisphere but overestimate it in the eastern hemisphere. The authors suggest a need to review and refine inventories and historical BC radiative forcing estimates, as those discrepancies would affect simulated surface temperatures and climate sensitivity, and thus would have implications for projections of future warming.

Considering the overall uncertainty related to BC in climate forcing estimates regionally and globally, the results and conclusions presented in this work are of significance. This work is original and the data presented support the conclusions, with no need for additional evidences, although there is no mention of a repository in which one could find the rBC data for the Flade Isblink and Akademii Nauk cores used in this study. All other data are referenced and available online.

I did not find inconsistencies in the manuscript or in the supplementary information, and consider the methodology applied robust and clearly explained. I do not see any impediments for this manuscript to be published as it is related to scientific merit, and have only two minor, specific comments.

Specific comments:

Reading the text and finding the cited figures was not fluent for me. It seems that the figures are not shown on the manuscript/supplement in the same order as they are mentioned. For example, Fig. S9 is mentioned in page 5 line 2, while Figure S1 is mentioned a few lines later (page 5 line 20). Another example: Figure 2 appears on page 4 line 5, but it is called only later, in page 5 line 2. I acknowledge, though, that this is more a personal opinion rather than a significant issue to be fixed.

There are two figures labeled Fig. S9 (in page 12 and on page 13 of the supplement), please check and also verify cross-reference along the manuscript and supplementary information.

Reviewer #2 (Remarks to the Author):

This paper entitled with “Revised historical black carbon emissions based on inverse modeling of ice core records” has tried to use the inverse model to simulate the BC emissions from the North America, Europe, Russia and the rest of Asia based on ice core BC records. Although, the modeling provided some useful data on the reconstructed emissions of BC during past 150 years. However, the manuscript currently failed to address the significance of this work, especially in the results and discussion. The authors showed the variations of simulated BC emission based on the ice core records, and discussed their potential sources, but failed to discuss why the original emission inventory has such differences with simulated ones. New points were focused on the peak or decreasing of BC emissions, why and what the implications for this modeling? Why there existed differences between ice core BC records based emission and CMIP used emission? Further, there are still many biases related to ice core dating, annual accumulation reconstruction, BC values under different measurement techniques, as well as model simulation. The last paragraph on the discussion of implication of BC, in recent years, there are many studies to estimate BC’s albedo effect on glacier melting, much more discussion is need to provide the new and profound understanding on BC historical emission on climate and snow/ice melting. Thus, I suggest substantial revision is needed to carefully address these comments.

1. In the abstract, the authors should show the time periods of ice core records here, making them

clear enough for readers to understand. TWO “with” in the line 26. Meanwhile, in the abstract, the authors should show the specific data you modeled, and briefly explain the reasons.

2. In the introduction, the authors should point out the major problems of BC emissions estimations currently and the merits of BC emission estimated from ice cores. What do we need to understand urgently? How to improve the understanding? What we will get from the estimation from ice core BC records?

3. When you discussed the BC records from lake sediment and ice cores in the introduction, you should notice that BC from lake sediments are not only from BC atmospheric deposition, but also from the river /runoff transport of BC from the lake drainage area. Ice core BC records could be from the atmospheric deposition, which can reflect the atmospheric BC transport and deposition. That may be possible reason to retrieve BC emission based on ice core records. However, precise dating ice cores are also important and difficult works.

4. In summary, the introduction need to be improved profoundly. More references on BC transport and its climate effect on climate and snow/ice melting should be added.

5. Page 3, line 3-5, I don't understand why you assume the BC atmospheric transport and deposition are the same since atmospheric circulation should be quite different in the different periods. Any model to simulate the BC transport and deposition in the past 200 years?

6. Page3, line 16, “...have lower ES values.”. What do you compared to? Compared to the surrounding regions, the BC ES in the northern Greenland is relatively higher. The reason you provided is quite confusing which was mainly caused by high altitudes?

7. Page 3, line 21, from the Figure 1c, ES is relatively higher over the Arctic Ocean, why? I believe the Arctic Ocean is not a BC source region.

8. Figure 2, how did you get the observation data of BC deposition from ice cores? Ice core records are proxies as common knowledge. What about their errors related to ice core dating and BC measurements?

9. Page 5, Line 5-6, how to eliminate the effects of different measurements of BC in ice core? How to uniform them in the comparable level?

10. Page 5, Line 9-11, how to explain the individual bias between each site? How to compare the modeled data with the observed data?

11. Page 6, Line 16, “decrease”? Underestimated?

12. Page 6, Line 27, you discussed the Southern Asia. However, from Figure 3d, I can see that the area covers Central Asia, East Asia and South Asia. It should be corrected. If the Everest ice core BC records (SP2 method) are included, we would see situation in Asia.

We thank the editor for handling the manuscript and the two reviewers for their comments. We answered the questions of the reviewers and followed their recommendation to add more details to both introduction and interpretation of the results. Additionally, we added references for all the datasets and models and uploaded the emission sensitivities for all ice cores in order to make our work reproducible and to enable other scientists to make use of our calculations for their applications. Below we repeat the comments/questions of the reviewers (*in italic*) and explain (bold font**) how we considered them in the new version of the manuscript (**red bold text, including page and line number, the same text is also inserted in red in the manuscript**):**

Reviewer #1 (Remarks to the Author):

Review of “Revised historical black carbon emissions based on inverse modeling of ice core records”

This work analyzes BC records from 13 ice cores from the Arctic, Europe and Asia and uses an inversion technique based on atmospheric transport and deposition modeling to reconstruct past BC emissions from North America, Europe, Russia and Asia. The results disagree with existing bottom-up inventories of BC emissions and imply that climate models using the CMIP5 and CMIP6 inventories underestimate BC radiative forcings in the western hemisphere but overestimate it in the eastern hemisphere. The authors suggest a need to review and refine inventories and historical BC radiative forcing estimates, as those discrepancies would affect simulated surface temperatures and climate sensitivity, and thus would have implications for projections of future warming.

Considering the overall uncertainty related to BC in climate forcing estimates regionally and globally, the results and conclusions presented in this work are of significance. This work is original and the data presented support the conclusions, with no need for additional evidences, although there is no mention of a repository in which one could find the rBC data for the Flade Isblink and Akademii Nauk cores used in this study. All other data are are referenced and available online.

I did not find inconsistencies in the manuscript or in the supplementary information, and consider the methodology applied robust and clearly explained. I do not see any impediments for this manuscript to be published as it is related to scientific merit, and have only two minor, specific comments.

We thank the reviewer for the positive overall assessment of our manuscript. With respect to the Flade Isblink and Akademii Nauk ice core data, we have posted the rBC fluxes in the Arctic Data Center repository (<https://doi.org/10.18739/A2W37KX21>) and updated the text accordingly.

Specific comments:

Reading the text and finding the cited figures was not fluent for me. It seems that the figures are not shown on the manuscript/supplement in the same order as they are mentioned. For example, Fig. S9 is mentioned in page 5 line 2, while Figure S1 is mentioned a few lines later (page 5 line 20). Another example: Figure 2 appears on page 4 line 5, but it is called only later, in page 5 line 2. I acknowledge, though, that this is more a personal opinion rather than a significant issue to be fixed.

There are two figures labeled Fig. S9 (in page 12 and on page 13 of the supplement), please check and also verify cross-reference along the manuscript and supplementary information.

Thank you for pointing out the duplicate figure label in the SI, we corrected this and checked all references to the figures. After re-reading we agree with your concerns and re-arranged the figures to agree better with the places where they are referred to in the text.

Reviewer #2 (Remarks to the Author):

This paper entitled with “Revised historical black carbon emissions based on inverse modeling of ice core records” has tried to use the inverse model to simulate the BC emissions from the North America, Europe, Russia and the rest of Asia based on ice core BC records. Although, the modeling provided some useful data on the reconstructed emissions of BC during past 150 years. However, the manuscript currently failed to address the significance of this work, especially in the results and discussion.

We think the work we performed is valuable to show that ice cores can be used for quantitative emission estimation and that our results could lead to the improvement of existing black carbon inventories by pointing out where more research is needed. We have added a substantial amount of text to address the reviewer’s concerns related to significance of the work.

The authors showed the variations of simulated BC emission based on the ice core records, and discussed their potential sources, but failed to discuss why the original emission inventory has such differences with simulated ones. New points were focused on the peak or decreasing of BC emissions, why and what the implications for this modeling? Why there existed differences between ice core BC records based emission and CMIP used emission?

The method used was designed to discover mismatches between existing bottom-up inventories widely used in Earth System Models and observations. Correcting the bottom-up inventories will require detailed historical and economic studies that are beyond the scope of this study and the expertise of our team. At several places we pointed out possible relationships with historical events such as the October Revolution in Russia and the Great Depression. However, it is impossible for us to say why the CMIP5 and CMIP6 data sets do not seem to reflect these events. The fact that the two inventories differ substantially - except during the last few decades - even though the methodologies used to develop them are quite closely related confirms large uncertainties in these inventories.

We added following in the text to stress the need for an update:

(Page 8, Line 14) “Thus, future efforts should be directed to further refining existing BC emissions inventories for the early 20th century to increase the reliability of climate models. Increased emissions in the early 20th century might also have impacted melting of Arctic ice due to albedo changes. Since our inversions suggest similar emissions during the Industrial Revolution and in the recent past, a higher melting rate seems possible compared to what climate models currently assume.”

Further, there are still many biases related to ice core dating, annual accumulation reconstruction, BC values under different measurement techniques, as well as model simulation.

We disagree with the reviewer about many of these issues for the cores we have selected for this study. Using parallel samples cut from the same ice core, we have shown that our SP2-based BC measurements in Greenland and Antarctic cores are highly repeatable (e.g., see McConnell et al., Nature, 2021). We also see a very consistent spatial pattern of change between Greenland BC records, particularly during since 1850 with the advent of widespread fossil fuel burning. In addition, most of the Greenland have dating uncertainties of less than a year during the last 150 years. While dating uncertainties in cores from other areas generally are higher, our study is based on decadal averages and so these dating uncertainties will have little impact. We also emphasize that all but one of the records used in our study were measured with the widely used SP2-based method so any uncertainties associated with different measurement techniques are limited. Text was added to clarify this.

Furthermore, we tested our results by eliminating individual ice cores and repeating the inversions. We obtained consistent results with respect to timing of the emission increments, confirming that the dating of the different ice cores is consistent with each other. In the paper, we have added somewhat more explanations to these issues:

(Page 3, Line 17) here we make use of an array of thirteen BC ice-core records collected from mid- to high-latitude Northern Hemispheric sites to constrain historical BC emissions since the late 19th century Industrial Revolution. BC deposition records were collected from Greenland, Arctic Canada, Russia, Svalbard, and mid-latitude glaciers in the European Alps and the Caucasus (supplementary Table S1). All the ice cores, except the Høltedahlfonna record, were analyzed using the single-particle soot-photometer method (McConnell et al., 2007b) so BC concentrations should be closely comparable, although there were differences in sample handling (e.g., continuous versus discrete sample measurements (Gleason et al., 2022) that may have affected some of the records. Temporal resolutions vary from sub-annual (Greenland) to sub-decadal (e.g., Canada); hence, we here focus on trends in decadal-scale deposition so the effects of any dating uncertainties, sampling resolutions, or post-depositional processes should be relatively small (e.g., Rose and Ruppel, 2015). The unprecedented size and geographic breadth of this ice-core array allows for exceptional robustness in the inferred BC emissions since BC deposition trends recorded in multiple ice cores likely reflect changes at regional to continental scales.

The last paragraph on the discussion of implication of BC, in recent years, there are many studies to estimate BC's albedo effect on glacier melting, much more discussion is need to provide the new and profound understanding on BC historical emission on climate and snow/ice melting. Thus, I suggest substantial revision is needed to carefully address these comments

To quantify the impact of BC on ice melting is a very important aspect and one of the key topics in studying Arctic BC. However, the focus of this paper was on determining the emissions, not the implications for ice melting. However, in the paper we have added the following statement:

(page 8, line 16) **“Increased emissions in the early 20th century might also have impacted melting of Arctic ice due to albedo changes. Since our inversions suggest similar emissions during the Industrial Revolution and in the recent past, a higher melting rate seems possible compared to what climate models currently assume.**

1. In the abstract, the authors should show the time periods of ice core records here, making them clear enough for readers to understand. TWO “with” in the line 26. Meanwhile, in the abstract, the authors should show the specific data you modeled, and briefly explain the reasons.

Thank you for spotting this, we removed one “with” and specified the ice cores used. We added the time period of the study to the abstract: “Here we use an objective inversion technique based on detailed atmospheric transport and deposition modeling to reconstruct 1850 to 2000 BC emissions from thirteen Northern Hemisphere ice-core records.”

2. In the introduction, the authors should point out the major problems of BC emissions estimations currently and the merits of BC emission estimated from ice cores. What do we need to understand urgently? How to improve the understanding? What we will get from the estimation from ice core BC records?

In the introduction, we wrote: “While the potential for climate change mitigation through BC emissions reductions is recognized, the development of successful mitigation strategies is hampered by significant uncertainties in bottom-up technology-based BC emission inventory data that are poorly constrained by observations².”

We have now added: “Therefore, more accurate quantification of BC emissions is fundamental for climate modelling. It is crucial to provide independent constraints on BC emissions, especially for historical emissions that are associated with the largest uncertainties because of a lack of observations.”

In the introduction, we also have a discussion about the use of ice cores for determining BC emissions. We have also added the statement: “Therefore we use a formal inversion technique to quantitatively constrain the emissions in several broad regions based on the ice core BC data.”

3a. When you discussed the BC records from lake sediment and ice cores in the introduction, you should notice that BC from lake sediments are not only from BC atmospheric deposition, but also from the river /runoff transport of BC from the lake drainage area.

Thank you for these improvement suggestions. Accordingly, we have made the following clarifications to the text:

(Page 2, Line 23): Long-term constraints on past BC emissions beyond the atmospheric monitoring era can be reconstructed from natural archives incorporated in lake sediments and glacier ice. While lake sediments are important archives of BC deposition since they are distributed throughout terrestrial regions, interpretation is hampered by low resolution, generally high dating uncertainties, and difficulties in quantitatively relating sediment records to atmospheric concentrations and emissions, in part because BC transport to lake sediment sites can be both aeolian and fluvial (Ruppel et al.,

2015, 2021). Contrary to sediment records, ice cores from the Greenland ice sheet and smaller mid- and high-latitude mountain glaciers and ice caps provide direct, well-resolved, and well-dated records of solely atmospherically deposited BC.

3b. Ice core BC records could be from the atmospheric deposition, which can reflect the atmospheric BC transport and deposition. That may be possible reason to retrieve BC emission based on ice core records. However, precise dating ice cores are also important and difficult works.

Yes, ice core dating is its own field of science and must be done carefully. However, ice cores directly archive material input in chronological order, and are not affected by vertical mixing such as sediments could be. The dating uncertainties are generally minor and do not affect the recorded long-term (decadal to centennial) trends in BC deposition in the ice cores. Furthermore, the dating uncertainties are insignificant on the temporal scale, i.e. decadal, on which the BC variations are discussed throughout the text and compared to the decadal modelling data.

4. In summary, the introduction need to be improved profoundly. More references on BC transport and its climate effect on climate and snow/ice melting should be added.

Indeed discussion on this are not strong enough in the paper, therefore we have substantially extended the introduction. More detailed information on the climate impact of BC and its effects on snow and ice melting is added throughout the Introduction. We also refer now to the newest results from the IPCC (AR6) report describing the atmospheric aerosol and the impact of snow deposition.

5. Page 3, line 3-5, I don't understand why you assume the BC atmospheric transport and deposition are the same since atmospheric circulation should be quite different in the different periods. Any model to simulate the BC transport and deposition in the past 200 years?

In fact the CERA-20C data set includes assimilated meteorological data since 1900. Thus, changes in meteorological conditions are fully captured in our transport model simulations on a 6-hourly basis. Prior to 1900, however, there are few re-analysis data sets available, and they are not very reliable – especially for the purpose of transport modelling and with respect to precipitation scavenging. Therefore, we used climatological information from CERA-20C to extend the meteorological record to 1850. However, sensitivity runs for the last 100 years show that using climatological data instead of actual meteorological data has little influence on our results. At least in the last 100 years, on decadal time scales, changes in BC emissions completely overwhelmed the effects of changes in meteorological conditions. As a result, decadal BC emissions in the last 100 years were not sensitive to using climatological or actual meteorological data. While we cannot test this for the period 1850-1900, we do not expect a major impact.

6. Page3, line 16, "...have lower ES values.". What do you compared to? Compared to the surrounding regions, the BC ES in the northern Greenland is relatively higher. The reason you provided is quite confusing which was mainly caused by high altitudes?

What we meant was that the air arriving at such high altitudes has very little contact with the surface compared to the other (lower-altitude) sites. Since emissions occur at (or near) the surface, high-altitude sites are less impacted by surface emissions. We have inserted the words “high-altitude” sites, to make it clearer that we refer to the ice core altitude.

7. Page 3, line 21, from the Figure 1c, ES is relatively higher over the Arctic Ocean, why? I believe the Arctic Ocean is not a BC source region.

ES is a measure of the potential for emission uptake, not of the emission contributions. Thus, ES can be high even in regions where there are no BC emissions. Thus, while the Arctic Ocean itself is not a source of BC, high ES values there tell us that the ice cores would be strongly impacted if emissions in this region increase (e.g., due to increases in shipping).

8. Figure 2, how did you get the observation data of BC deposition from ice cores? Ice core records are proxies as common knowledge. What about their errors related to ice core dating and BC measurements?

Thank you for pointing out that we had inadvertently forgotten to describe how the depositional BC fluxes had been calculated from the observed BC concentrations in the ice cores. We have added the following text to the Methods section “BC Measurements”:

(page 9, line 26) “For each ice core, following standard procedures, annual the BC deposition (mg m⁻² yr⁻¹) was calculated from the observed BC concentrations by multiplying by the annual net water-equivalent accumulation. For Holtedahlfonna, EC deposition was calculated by dividing the quantified total amount of EC on a filter sample by the known cross section of the ice sample and the number of years covered in one filtered ice sample (Ruppel et al. 2014).”

Yes, potential errors in dating of the ice cores affect the BC deposition calculations and thus we added on lines the following:

(page 9, line 20) Potential sources of uncertainties in the BC deposition fluxes include instrumental limitations (e.g., under-detection of very large BC particles), dating errors (< ±5 years in some cores), post-depositional effects (e.g., wind scouring of snow, melt-refreeze cycles) and spatial noise (variability in deposition across space). For cores collected in the dry firn zone of Greenland, these sources of uncertainties are relatively minor, but some are poorly quantified at other coring sites, and will require further study. However, it is unlikely that such uncertainties would affect BC deposition trends on decadal scales that are the focus of the present study.

9. Page 5, Line 5-6, how to eliminate the effects of different measurements of BC in ice core? How to uniform them in the comparable level?

The differences in magnitudes between observed and modeled BC deposition are discussed thoroughly in the Supporting Information section “Temporal variation of BC emissions related to measured and modeled BC deposition in ice cores”.

For the inversion, we scaled the modeled values to the observations over the most recent decade. **This emphasizes the temporal variability and removes** uncertainties associated with absolute BC concentrations measured using different methods.

10. Page 5, Line 9-11, how to explain the individual bias between each site? How to compare the modeled data with the observed data?

Differences in BC deposition and trends between individual ice coring sites are expected for the reasons mentioned above (different proximity to emission sources and variable atmospheric transport of BC to the glaciers). As stated earlier, the differences in magnitudes between observed and modeled BC deposition are discussed thoroughly in the Supporting Information section “Temporal variation of BC emissions related to measured and modeled BC deposition in ice cores”. Note that ice cores that are relatively close to each other have very similar deposition fluxes, indicating the ice core measurements are robust.

11. Page 6, Line 16, “decrease”? Underestimated?

The result of the inversion are updated emissions, therefore the term “decrease” is correct. If the results were compared to observations the term “underestimated” would be suitable.

12. Page 6, Line 27, you discussed the Southern Asia. However, from Figure 3d, I can see that the area covers Central Asia, East Asia and South Asia. It should be corrected. If the Everest ice core BC records (SP2 method) are included, we would see situation in Asia.

We corrected the description from Asia to CES-Asia and clarified that this is Central, Eastern and Southern Asia in the figure caption. We acknowledge that the Everest and Tibetan plateau ice cores are an important indicator of Asia's atmospheric BC variability, particularly for South Asia. However, in this paper we focus on mid to high latitudes where anthropogenic emissions historically have been highest. We fully agree that it would be interesting to investigate South Asia as well but this would deserve a dedicated study on its own, ideally using several ice cores from CES-Asia. In this way a targeted study of Asian emissions of all latitudes could be performed. This would give a better emission representation over all Asia.

REVIEWER COMMENTS

Reviewer #2 (Remarks to the Author):

I am happy to see the manuscript has been greatly improved after the revision. The authors have replied most of my previous concerns. It is a pity that the authors didn't include BC ice core records in the Tibetan Plateau where BC has accelerated glacier and snow cover melting. I accept their explanation for focusing on mid-high latitudes of the Northern Hemisphere in current work. I have some minor comments for further considering.

Since the work focuses on the Northern Hemisphere, the title should be changed to "Revised historical black carbon emissions based on inverse modeling of ice core records in the Northern Hemisphere".

Page 2 Line 9, please check updated knowledge and understanding, such as:
Kang et al., 2020, A review of black carbon in snow and ice and its impacts on cryospheric change. Earth-Science Reviews, 210, 103346. <https://doi.org/10.1016/j.earscirev.2020.103346>
Kang et al., 2019. Linking Atmospheric Pollution to Cryospheric Change in the Third Pole Region: Current Progresses and Future Prospects. National Science Review, 6(4): 796-809. <https://doi.org/10.1093/nsr/nwz031>.

Page 2 Line 15, please use "South Asia" which is a specific regional term.

Page 2 Line 18, please add since when the BC pollution in North America and Europe has been decreasing?

Page 2, several sentences lack of comma, for example, line 6, line 8, line 20, line 29, line 34.

Page 3 Lines 28-30, For the FES, is BC emission uptake the same to BC deposition? if not, what are their differences? if they are same, why use uptake here?

Page 4 Figure 1, why the authors use ES here, not the FES? What are the differences between ES and FES?

Page 6 Line 16, please use East Asia and South Asia.

Reply to Reviewer 2

The authors thank reviewer 2 for another checkup on the paper and the improvements he suggested to us.

I am happy to see the manuscript has been greatly improved after the revision. The authors have replied most of my previous concerns. It is a pity that the authors didn't include BC ice core records in the Tibetan Plateau where BC has accelerated glacier and snow cover melting. I accept their explanation for focusing on mid-high latitudes of the Northern Hemisphere in current work. I have some minor comments for further considering.

Since the work focuses on the Northern Hemisphere, the title should be changed to "Revised historical black carbon emissions based on inverse modeling of ice core records in the Northern Hemisphere".

I understand your suggestion for a change in the title, but there is a limit of 15 words, so I cannot accommodate the four additional words "In the Northern Hemisphere". Maybe the editor could allow an exception.

Page 2 Line 9, please check updated knowledge and understanding, such as:

Kang et al., 2020, A review of black carbon in snow and ice and its impacts on cryospheric change. Earth-Science Reviews, 210, 103346. <https://doi.org/10.1016/j.earscirev.2020.103346>

Kang et al., 2019. Linking Atmospheric Pollution to Cryospheric Change in the Third Pole Region: Current Progresses and Future Prospects. National Science Review, 6(4): 796-809. <https://doi.org/10.1093/nsr/nwz031>.

Thank you for suggesting this literature indeed it is valuable to also point the reader to the third pole region, we added the references.

Page 2 Line 15, please use "South Asia" which is a specific regional term

Done.

Page 2 Line 18, please add since when the BC pollution in North America and Europe has been decreasing?

We added "in the last decades"

Page 2, several sentences lack of comma, for example, line 6, line 8, line 20, line 29, line 34.

We added the commas and had a native speaker look over it.

Page 3 Lines 28-30, For the FES, is BC emission uptake the same to BC deposition? if not, what are their differences? if they are same, why use uptake here?

Indeed the term uptake might be confusing here, we reformulated the paragraph to:

“For each ice-core site we calculated monthly footprint emission sensitivities (FES) that allow quantification of the impact of emissions on the BC deposition at the receptor. FES is the probability that emissions occurring at any gridcell lead to deposition at the receptor. Thus, FES can be high even in regions where there are no BC emissions.”

Page 4 Figure 1, why the authors use ES here, not the FES? What are the differences between ES and FES?

The term FES was introduced later to substitute ES, but not at all instances, we changed now all ES to FES.

Page 6 Line 16, please use East Asia and South Asia.

done